

# Winter atmospheric boundary layer observations over sea ice in the coastal zone of the Bothnian Bay (Baltic Sea)

Marta Wenta[1], David Brus[2], Konstantinos Doulgeris[2], Ville Vakkari[2], and Agnieszka Herman[1]

[1]University of Gdansk, Institute of Oceanography, al. Marszałka Piłsudskiego 46, 81 - 378 Gdynia, Poland
[2]Finnish Meteorological Institute, FI-00101 Helsinki, Finland

**Correspondence:** Marta Wenta (marta.wenta@phdstud.ug.edu.pl)

**Abstract.** The Hailuoto Atmospheric Observations over Sea ice (HAOS) campaign took place at the westernmost point of Hailuoto island (Finland) between 27 February and 2 March 2020. The aim of the campaign was to obtain atmospheric boundary layer (ABL) observations over seasonal sea ice in the Bay of Bothnia. Throughout 4 days both fixed-wing and quad-propeller rotorcraft unmanned aerial vehicles (UAV) were deployed over the sea ice to measure the properties of the lower

ABL and to obtain accompanying high-resolution aerial photographs of the underlying ice surface. Additionally, a 3D sonic anemometer, an automatic weather station and a Halo Doppler lidar were installed on the shore to collect meteorological observations. During the UAV flights, measurements of temperature, relative humidity and atmospheric pressure were collected at 4 different altitudes between 25 m and 100 m, over an area of $\sim$1.5 km$^2$ of sea ice, located 1.1–1.3 km offshore from the Hailuoto Marjaniemi pier, together with orthomosaic maps of the ice surface below. Altogether the obtained dataset consists of

27 meteorological flights, 4 photogrammetry missions and continuous measurements of atmospheric properties from ground-based stations located at the coast. The acquired observations have been quality controlled and post processed and are available through the PANGAEA repository (https://doi.pangaea.de/10.1594/PANGAEA.918823, Wenta et al., 2020).

## 1 Introduction

Small-scale processes at the atmosphere–sea ice–ocean interface are considered crucial to improve the performance of numerical weather prediction (NWP) models for the polar regions (Vihma et al., 2014). Sea ice, due to its low conductivity, isolates the ocean from the atmosphere and blocks the exchange of heat and moisture. However, due to wind forces, ocean currents and internal pressure, the sea ice surface is not homogeneous but covered with ridges, cracks and leads. All those features, in particular areas of open water or very thin ice, affect the properties of the overlying atmospheric boundary layer (ABL) both

locally and regionally (e.g., Manucharyan and Thompson, 2017; Wenta and Herman, 2018, 2019; Batrak and Müller, 2018) and play an important role in sea ice dynamics and the evolution of seasonal sea ice extent (e.g., Horvat and Tziperman, 2015; Zhang et al., 2018). In situ observations of ABL properties over sea ice are essential for expanding our knowledge about phys-





ical processes at the interface of the ocean, sea ice and atmosphere and for the development of parameterizations necessary for improvement of NWP models.

For many years, observations of the ABL over inhomogenous sea ice have focused on satellite remote sensing (e.g., Qu et al., 2019), manned aircraft (e.g., Frech and Jochum, 1999; Brümmer, 1999; Tetzlaff et al., 2015) and expensive field campaigns (LEADEX, SHEBA; LeadEx Group, 1993; Uttal et al., 2002). While those data sources considerably increased our understanding of sea ice–atmosphere interactions (Vihma et al., 2014), there are still many gaps in the observations of the submesoscale processes at the interface of sea ice and the ABL. An approach that allows to overcome many of the shortcomings of earlier

field campaigns in the polar regions are unmanned aerial vehicle (UAV) operations. The usage of UAVs in harsh conditions associated with cryospheric studies has been continuously increasing throughout the last 15 years (Gaffey and Bhardwaj, 2020; Bhardwaj et al., 2016), as they provide an opportunity to reach previously inaccessible areas and to obtain three-dimensional observations of the ABL. Formerly such measurements were either impossible or too expensive and required several measuring platforms instead of one. The ABL and sea ice properties have already been a subject of several UAV campaigns focusing on

the marginal sea ice zone (MIZOPEX; Zaugg et al., 2013), polynyas (Knuth. et al., 2013; Cassano et al., 2015) and ABL structure offshore (deBoer et al., 2018). Another relevant campaign employing UAVs for the observations of stable atmospheric boundary layer over sea ice is "Innovative Strategies for Observations in the Arctic Atmospheric Boundary Layer" (ISOBAR) (Kral et al., 2018) which took place at and off the coast of Hailuoto island (Finland), i.e., the location of the present study.

The main goal of HAOS was to study the ABL response to sea ice surface inhomogeneities at different times of the day. Due

to a very warm winter 2019/2020 and the associated exceptionally small sea ice extent in the Bay of Bothnia (and in the Baltic Sea in general) in the first months of 2020, the number of potential locations that would fit the purpose of our research was very limited. Eventually, after monitoring of the development of weather and sea ice conditions throughout February 2020, the westernmost point of the Finnish island Hailuoto (Fig. 1c), around a small harbour of Marjaniemi, was chosen as the most suitable location. The island is situated ~20km from the city of Oulu, in the north-eastern part of the Bay of Bothnia. During

the period of interest, the waters surrounding Hailuoto were covered with landfast ice, as is typical for this region in February. Importantly, throughout the first two months of 2020 the edge of the sea ice cover was located only a few hundred meters off the coast. Consequently, the drifting ice pack that developed seawards from the landfast-ice zone at the end of February, interesting from the point of view of the HAOS campaign, was within reach of our UAVs (Fig. 2).

Between February 27th and March 2nd 2020, a series of UAV flights were undertaken off the Marjaniemi harbour (Fig. 1a,b),

accompanied by continuous ground-based observations of the lower atmosphere. Two different small UAVs were used: a fixed-wing UAV (called UAV-UG1) and a multi-rotor DJI Mavic 2 Pro. Apart from initial tests on 27 February, a total number of 23 UAV-UG1 flights took place, each covering the same area of 1.37×1.1 km, located 1.3~1.1 km from the starting/landing point at the Hailuoto Marjaniemi pier. The second, multi-rotor drone took overlapping aerial images of sea ice over the same area, which were later used to create orthomosaic maps. In addition, a meteorological station and a Halo Doppler lidar instrument

were installed at the pier (position: 65.039684°N, 24.555065°E), and collected data throughout the whole campaign. A detailed description of all instruments and the measurement methodology are provided in the following sections.

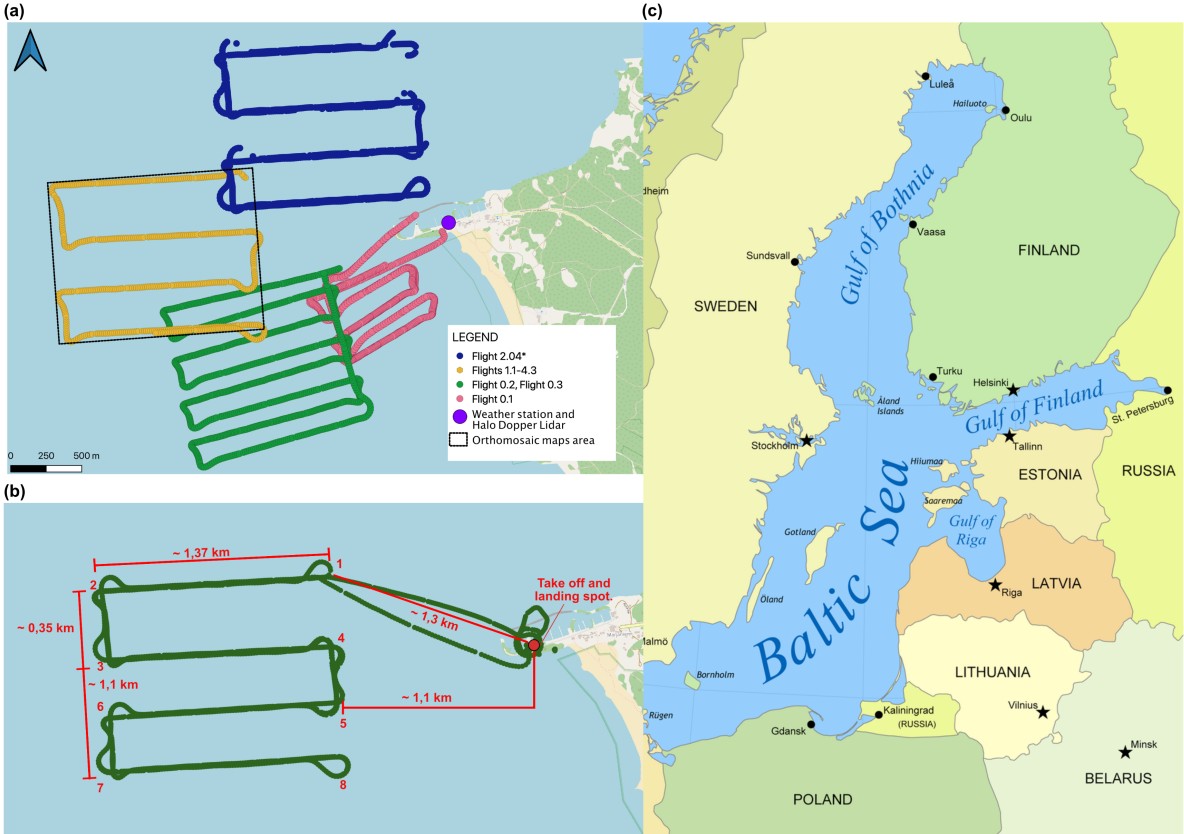

**Figure 1.** (a) Flight paths during the HAOS campaign, flight labels as in Table 1. (b) The flight path of the meteorological measurement mission (Tab. 1, Flights 1.1–4.3) with distances from the take-off and landing spot close to Marjaniemi lighthouse pier. (c) The Baltic Sea with the highlighted location of Hailuoto.

## 2   UAV-based measurements

### 2.1   sUAV meteorological profiling

The small unmanned aerial vehicle (sUAV) platforms (UAV-UG1 and UAV-UG2, UG meaning the University of Gdansk) used during HAOS campaign were built around the ZOHD nano Talon fixed-wing V-tail airframe (Fig. 3a). The sUAV was developed at the Finnish Meteorological Institute (FMI) as an inexpensive measurement platform to operate in a variety of conditions. The nano Talon is a small pusher-propeller aircraft, with 860 mm wingspan and all-up weight less than 1.5 kg. The maximum endurance of these aircrafts is about 60 min using 16.8 V, 3200 mAh rechargeable lithium-ion (Li-ion) batteries. Flights were carried out using a flight controller (Matek F405 wing) with the Ardupilot software. The propulsion system, consisting of 1870 kV brushless motors, 30A Electronic Speed Controllers and 6 inch (3 inch pitch) propeller, was used for both rotorcraft. In HAOS, all flights were conducted with UAV-UG1 platform, having UAV-UG2 as spare one. The ground



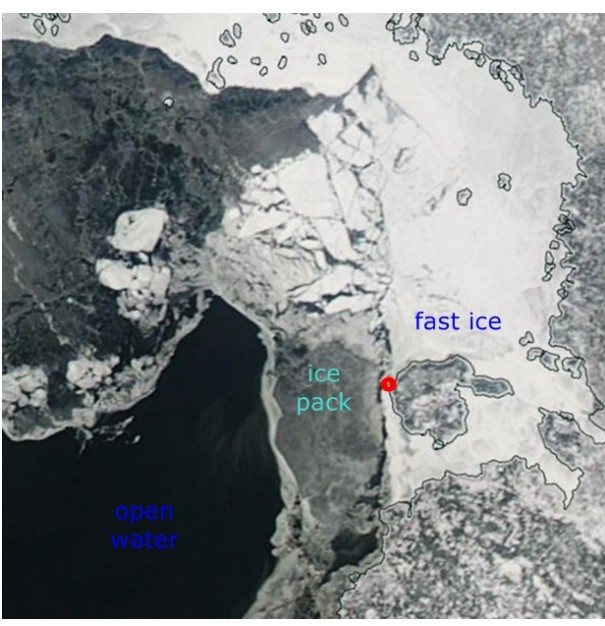

**Figure 2.** Sea ice conditions in the inner parts of the Bay of Bothnia on Feb 28th 2020 (Day 1 of the campaign): MODIS Aqua image with marked fast ice, ice pack and open water areas. The red dot shows the location of the Marjaniemi harbour.

radio controller and UAV communicated via 868 MHz radio frequency with a range of more than 10 km. The aircraft utilized a first-person viewer (FPV) video link at 5.8 GHz, which enabled visual monitoring of the UAV performance with real-time on-screen-display (OSD) telemetry (Fig. 3b). All flights during HAOS campaign were completed as autopilot guided missions except for take–off and landing operation, when they were under the control of an operator.

The platform carried a pair of meteorological sensors (Bosh BME280, P (hPa), T (°C) and RH (%)) for measurements of the atmospheric state and the redundant GPS unit (lon (deg), lat (deg), alt (m, $\pm$ Mean Sea Level)), both connected to Raspberry Pi zero W. The BME280 sensors were attached on each side of the aircraft fuselage under each wing (Fig. 3a). The attachment of the sensors was done via 3D-printed housing with the distance from the fuselage about 1.5 cm, allowing free airflow around the sensor and shielding it from the solar radiation. The BME280 sensor has a manufacturer-stated response time and accuracy of 6 ms, $\pm$1 hPa for pressure, 1 s and $\pm$0.5 °C for temperature, and 1 s, $\pm$3% RH for relative humidity. The BME280 sensors were T and RH calibrated (both 6 points) at the FMI Observation Unit against the national standard in the range of $-20$°C < T < 20°C and 25% < RH < 94% at 10°C.

The platform obtained measurements at high spatial resolution with the average flight cruising speed of about 12 m s$^{-1}$ and burst up to 25 m s$^{-1}$. The flights were performed in two cycles (morning and afternoon) with 2-4 flights in each cycle and about 45 minutes between flights (Table 1). The measured data were logged at a rate of 1 Hz as ASCII comma-separated-variables files (csv) to an embedded Raspberry Pi zero W minicomputer using simple Python scripts. The signals from the meteorological sensors and from the GPS were aligned in time during post-processing, using cross-correlation techniques. Data preprocessing


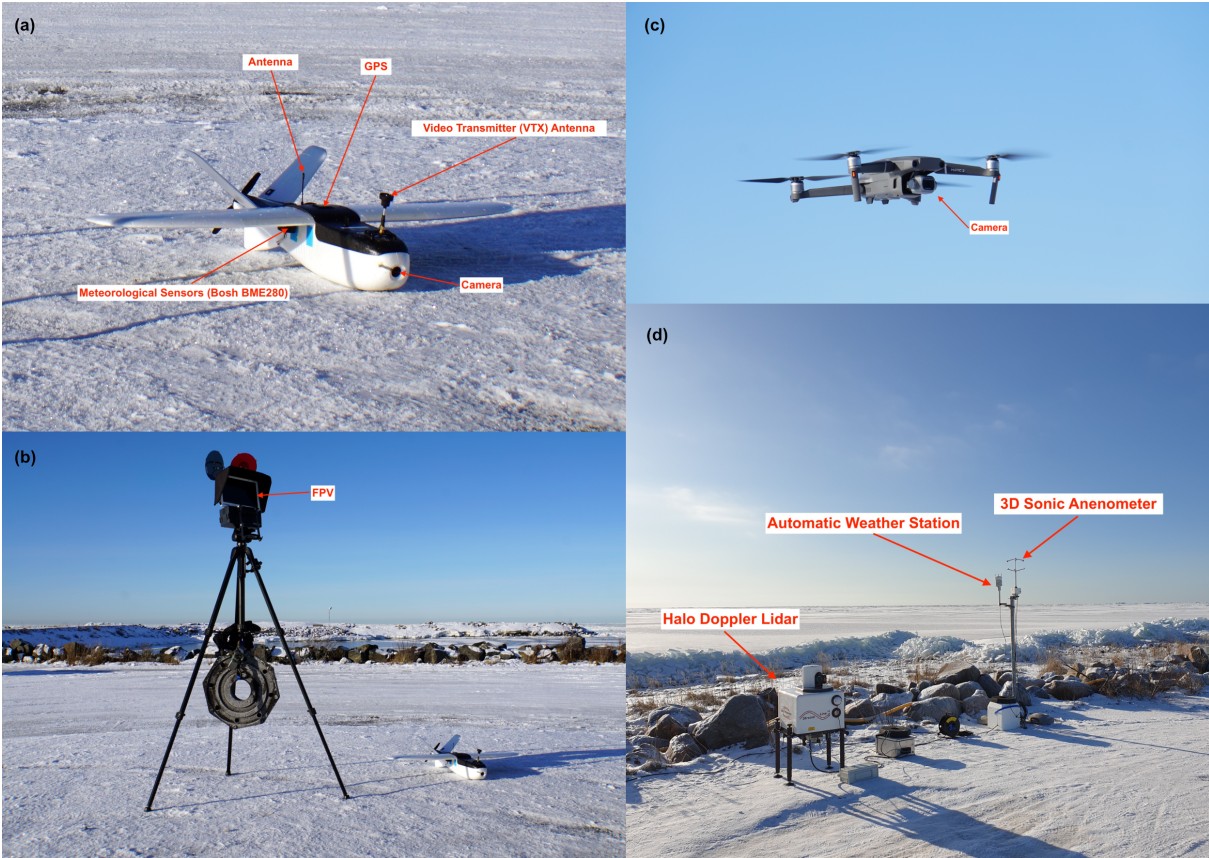

**Figure 3.** (a) UAV-UG1. (b) Telemetry screen for FPV real-time on-screen display of the UAV-UG1 performance. (c) DJI Mavic 2 Pro during flight. (d) Ground meteorological measurements.

included also the removal of the initial ("to") and final ("back") segments from each flight, i.e., before and after the sUAV
85  reached its prescribed path – see further section 2.3.

## 2.2  Photogrammetry sUAV

Aerial photography of the sea ice was done using DJI Mavic 2 Pro consumer-orientated quad-propeller rotorcraft with AUW
of 907 g and 354 mm rotor-to-rotor distance (Fig. 3c). The maximum manufacturer-stated endurance of the rotorcraft is about
30 minutes (under no-wind conditions) using nominal 15.4 V, 3850 mAh rechargeable Lithium-Polymer (Li-Po) batteries.
90  The rotorcraft communicated with ground DJI radio remote controller via 2.4 GHz frequency with max transmission distance
of 5 km. The rotorcraft utilized a first-person viewer (FPV) video link proprietary DJI OcuSync 2.0 system with real-time
on-screen-display (OSD) telemetry on Android-based mobile device connected via USB port to the remote controller. The
rotorcraft is equipped with a 3-axis stabilized gimbal holding a camera with 1-inch CMOS sensor having 20 million effective
pixels (5472×3648 px.), a lens with the field-of-view (FOV) of about 77 deg and aperture range of f/2.8–f/11.





**Table 1.** Meteorological flights performed by UAV-UG1 with minimum and maximum values of measured parameters.

| Date | Flight Number | Start Time (UTC+2) | End Time (UTC+2) | Duration (hh:mm:ss) | min T (°C) | min P (hPa) | min RH (%) | max T (°C) | max P (hPa) | max RH (%) |
|---|---|---|---|---|---|---|---|---|---|---|
| Day 0 – 27.02.2020 (tests) | Flight 0.1 | 15:34:42 | 15:59:49 | 00:25:07 | -2.44 | 989.62 | 75.59 | -1.16 | 1002.79 | 80.54 |
| | Flight 0.2 | 16:17:23 | 16:32:02 | 00:14:39 | -3.22 | 994.40 | 71.26 | -0.66 | 1002.85 | 80.44 |
| | Flight 0.3 | 17:36:53 | 17:53:15 | 00:16:22 | -4.54 | 990.06 | 75.12 | -2.00 | 1003.05 | 84.14 |
| Day 1 – 28.02.2020 | Flight 1.1 | 10:33:28 | 11:05:39 | 00:32:11 | -8.53 | 992.16 | 39.76 | -0.22 | 1004.85 | 72.35 |
| | Flight 1.2 | 11:37:56 | 12:10:38 | 00:32:42 | -7.31 | 992.30 | 54.14 | -2.97 | 1005.09 | 74.11 |
| | Flight 1.3 | 15:58:18 | 16:29:21 | 00:31:03 | -4.02 | 991.78 | 33.57 | -0.07 | 1004.32 | 58.81 |
| | Flight 1.4 | 16:40:41 | 17:10:24 | 00:29:43 | -4.59 | 991.56 | 38.20 | -2.69 | 1004.14 | 65.09 |
| | Flight 1.5 | 17:23:15 | 17:54:13 | 00:30:58 | -5.80 | 991.64 | 43.43 | -3.41 | 1004.08 | 64.94 |
| Day 2 – 29.02.2020 | Flight 2.1 | 09:12:02 | 09:44:09 | 00:32:07 | -10.49 | 986.28 | 41.69 | -3.58 | 999.23 | 83.36 |
| | Flight 2.2 | 10:07:26 | 10:39:39 | 00:32:13 | -9.06 | 986.14 | 57.95 | -4.48 | 999.28 | 81.13 |
| | Flight 2.3 | 11:56:30 | 12:28:54 | 00:32:24 | -6.76 | 985.7 | 51.97 | -4.94 | 998.6 | 70.27 |
| | Flight 2.4* | 13:01:46 | 13:33:17 | 00:31:31 | -5.76 | 985.08 | 43.49 | -3.56 | 998.08 | 59.52 |
| | Flight 2.5 | 15:28:27 | 16:00:15 | 00:31:48 | -3.69 | 984.74 | 32.87 | 7.14 | 997.08 | 65.75 |
| | Flight 2.6 | 16:12:45 | 16:43:44 | 00:30:59 | -4.03 | 984.65 | 53.9 | -0.51 | 997.00 | 62.59 |
| | Flight 2.7 | 16:53:25 | 17:24:56 | 00:31:31 | -5.43 | 984.44 | 52.39 | -2.60 | 996.86 | 66.80 |
| | Flight 2.8 | 17:34:02 | 18:07:06 | 00:33:04 | -6.42 | 984.19 | 51.37 | -2.77 | 996.88 | 63.49 |
| Day 3 – 01.03.2020 | Flight 3.1 | 09:32:37 | 10:05:13 | 00:32:36 | -5.08 | 978.67 | 68.21 | -2.89 | 991.8 | 78.59 |
| | Flight 3.2 | 10:15:54 | 10:47:22 | 00:31:28 | -4.96 | 979.13 | 74.75 | -3.55 | 991.83 | 80.04 |
| | Flight 3.3 | 11:59:18 | 12:30:47 | 00:31:29 | -4.86 | 979.53 | 33.65 | 9.12 | 992.02 | 79.58 |
| | Flight 3.4 | 12:42:41 | 13:13:17 | 00:30:36 | -4.91 | 979.31 | 72.73 | -2.38 | 991.87 | 82.72 |
| | Flight 3.5 | 15:28:28 | 16:02:20 | 00:33:52 | -4.04 | 978.55 | 34.44 | -1.29 | 991.21 | 72.92 |
| | Flight 3.6 | 16:12:28 | 16:45:03 | 00:32:35 | -4.58 | 978.76 | 68.93 | -2.92 | 991.1 | 73.98 |
| | Flight 3.7 | 16:54:51 | 17:27:28 | 00:32:37 | -4.89 | 978.61 | 70.2 | -2.99 | 991.22 | 76.31 |
| | Flight 3.8 | 17:37:10 | 18:10:39 | 00:33:29 | -5.12 | 978.27 | 69.93 | -3.80 | 991.23 | 76.13 |
| Day 4 – 02.03.2020 | Flight 4.1 | 09:21:28 | 09:54:27 | 00:32:59 | -13.38 | 984.55 | 49.52 | -8.03 | 998.49 | 80.19 |
| | Flight 4.2 | 10:06:36 | 10:45:30 | 00:38:54 | -13.48 | 985.24 | 74.27 | -10.92 | 998.94 | 80.23 |
| | Flight 4.3 | 10:58:00 | 11:32:20 | 00:34:20 | -13.88 | 985.74 | 73.46 | -11.43 | 999.68 | 84.00 |

## 2.3 Mission planning

Two separate missions were planned for meteorological measurements and photogrammetry sea ice surface mapping. The meteorological measurements mission planning was done using the Mission Planner software. As already mentioned, the survey area was ~1.37 km long and ~1.1 km wide, i.e., it covered ~1.5 km$^2$. The route design comprised flights at four altitudes: 25, 50, 75 and 100 m above ground level (AGL) as a zigzag line with 3 main turns with distance of ~0.35 km between the legs (see Fig. 1b, Fig. 4). The aircraft flew at a constant altitude through the first way point (No. 1 in Fig. 1b), positioned at the upper right-hand corner, and then followed the serpentine pattern to the last way point at the lower right-hand corner (No. 8 in Fig 1b), where the aircraft turned 180 degrees and started to climb from way point No. 8 back to the way point No. 7 reaching the next mission altitude, which was followed in the opposite direction to the lower one, i.e., it was completed when the aircraft reached again the starting way point No. 1. The whole procedure repeated till the programmed mission was completed, i.e., the aircraft reached the last waypoint, No. 1, at the attitude of 100 m. At this point, the aircraft switched to return-to-launch flight mode.

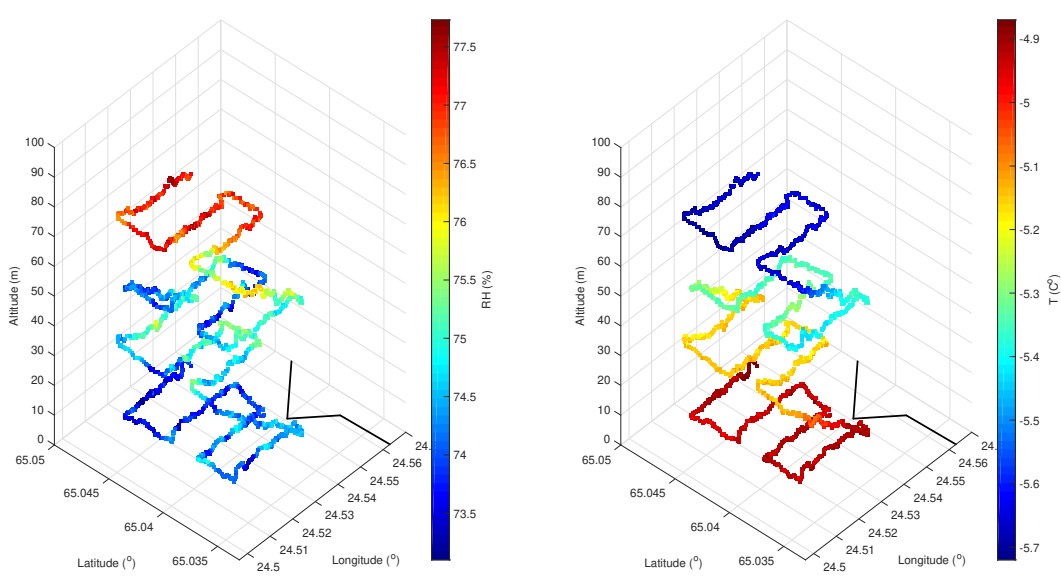

**Figure 4.** UAV-UG1 measurements of (a) relative humidity and (b) air temperature from Flight 3.2 (Tab.1). Black line indicates Hailuoto Marjaniemi shoreline.

Besides the missions described above, the meteorological measurements included also test flights on Day 0 (Tab. 1, Fig. 1, Flight 0.1-0.3) and an additional flight launched on 29 February 2020 (Tab. 1, Fig. 1 Flight 2.4*). Flight 2.4* took place over the area of PILOT boat L144 passage, with an aim to investigate whether the modification of the ice surface along the path of 110 that boat affected the atmospheric properties above. The shape of the path of this "additional" mission was identical to that of the Flights 1.1-4.3, but located in a different area, as shown in Fig. 1a.

The photogrammetry mission planning was done using the Android-based Pix4Dcapture (version 4.8.0) application as a grid mission. The survey area was the same as for the meteorological missions, but it was divided into four separate, vertically overlapping segments of $0.4 \times 1.1$ km. The flight altitude was set to 150 m AGL with ground sampling distance (GSD) of 115 3.3 cm·px$^{-1}$. The pictures overlap rate at both sides equaled 80% and the camera angle was set to 90 degrees. Each of the four flights necessary to cover the whole survey area lasted about 20 min and the battery had to be changed between the flights. Every day of the campaign, except on Day 4 (March 2nd), one aerial photography mission was performed with the number of collected images equal to 392 (testing missions), 970, 1144 and 1171 on Day 0 to Day 3, respectively. The areas covered equaled 0.301 km$^2$, 1.561 km$^2$, 1.643 km$^2$ and 1.241 km$^2$. The flights were performed under sunny and partially cloudy, 120 cold and moderate wind weather conditions. Importantly, no clouds were present between the aircraft and the surface. The low-resolution, overview pictures of all four orthomosaic sea ice maps are presented in Fig. 5.



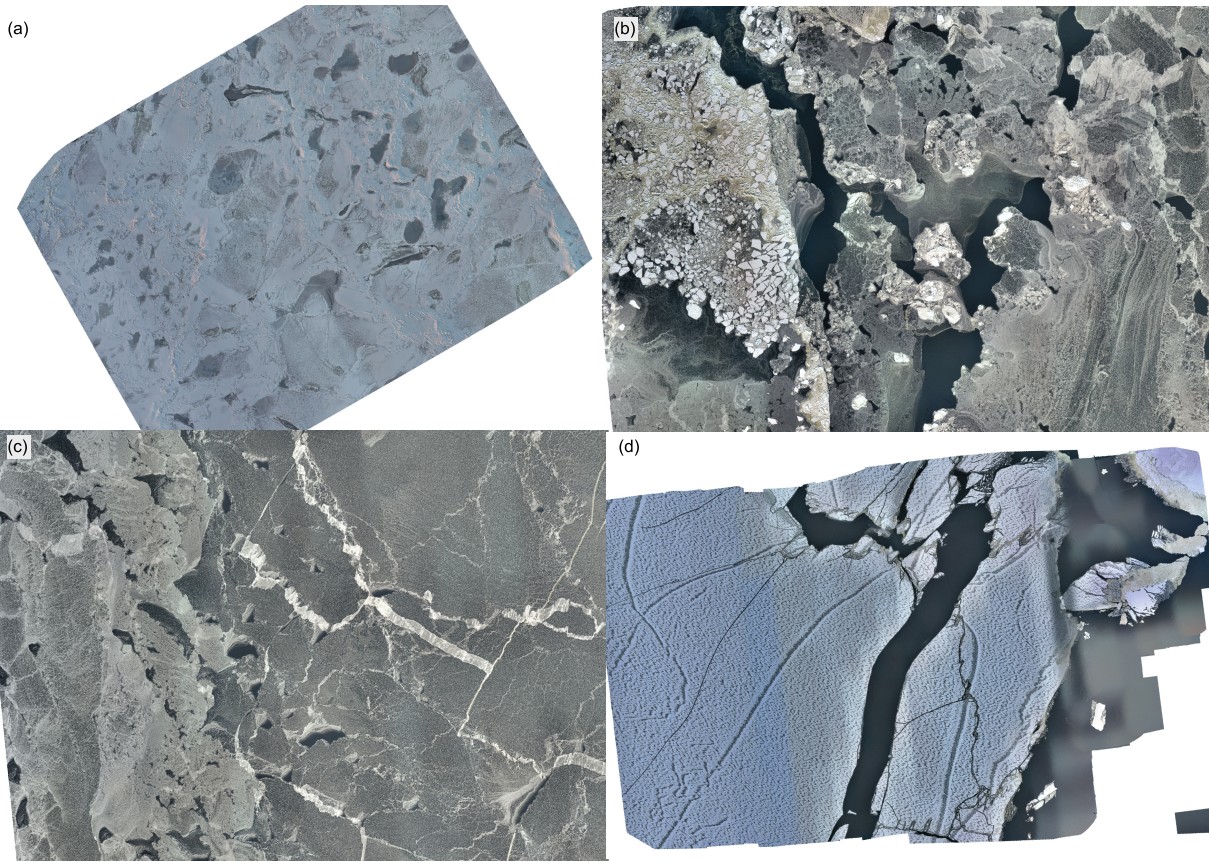

**Figure 5.** Overview of the orthomosaic maps of the ice surface generated by stitching individual images from DJI Mavic 2 Pro photogrammetry missions: (a) first map from 27 February, located ∼65 m to the south of Hailuoto Marjaniemi pier; maps of the main survey area (Fig. 1b) from (b) 28 February, (c) 29 February and (d) 1 March.

## 2.4 Image Processing

The standalone version of Pix4Dmapper software version 4.5.6 was used to process the collected images. The rotorcraft camera was calibrated automatically as a part of the SfM process by Pix4D mapper software. During our aerial photography missions
no ground control points (GCP) were used, since the logistics on the sea ice sheet was impossible due to many cracks and very thin ice. Our interest was only in generating the orthomosaic overlays GeoTiff and Google Maps tiles and KML files in WGS84 (EGM 96 Geoid) Coordinate System to facilitate superposition of the meteorological data and and sea ice maps for a subsequent analysis. The following processing settings were used: keypoints image scale: full; image scale: 1; point cloud densification image scale: multiscale, 1/2 (half image size); point density: optimal; minimum number of matches: 3; matching
image pairs: aerial grid or corridor; targeted number of key points: automatic, rematch: automatic; 3D textured mesh; medium

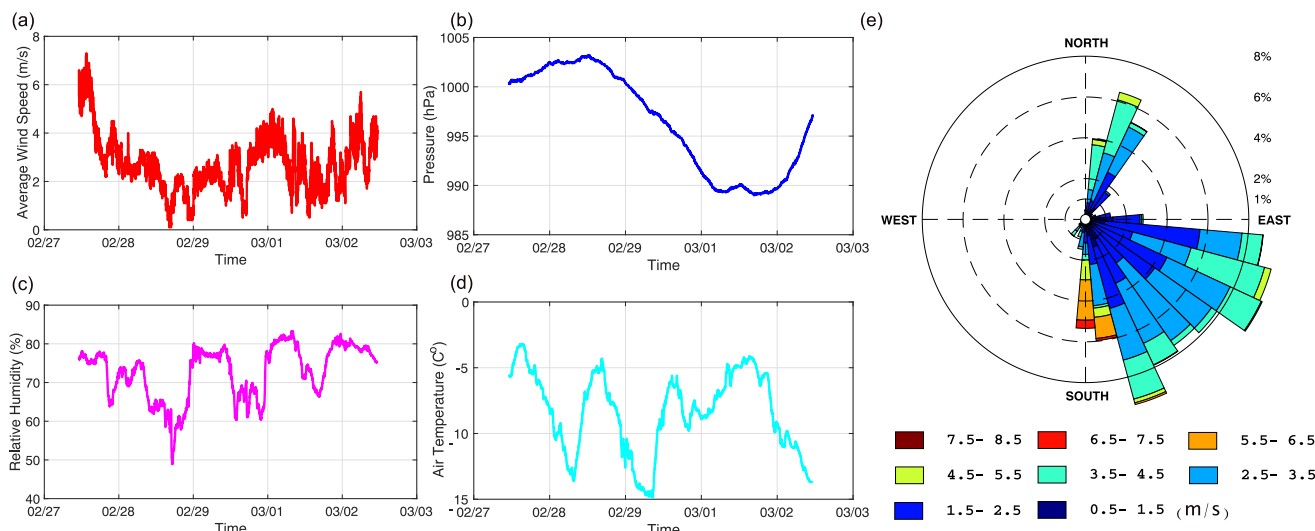

**Figure 6.** Automatic weather station measurements during the HAOS campaign: time series of (a) wind speed, (b) atmospheric pressure, (c) relative humidity, (d) air temperature, and (e) statistics of wind direction and speed.

resolution, no color balancing. The following hardware was used: CPU: Intel(R) Core(TM) i7-8750H CPU with 2.20GHz, RAM: 64GB, GPU Intel(R) UHD Graphics 630, NVIDIA Quadro P1000, operating system Windows 10 Pro, 64-bit.

## 3 Ground-Based Measurements

### 3.1 Ground Meteorological Measurements

The ground meteorological observations were done by a 3D sonic anemometer (uSonic-3 Scientific (former: USA-1), METEK GmbH) and an automatic weather station (WXT, Vaisala Inc.). Both instruments were mounted on a metal mast at the height of 2.5 m above the ground surface (Fig. 3d). The 3D anemometer measured 3 wind speed components (u, v, w in $m·s^{-1}$) and acoustic temperature (T in degr C) at 10 Hz resolution. The Vaisala WXT sensor measured the ambient temperature, relative humidity, rain intensity, wind direction and wind speed. The following parameters were logged as a 1-minute averages: date

and time (DD-MM-YY HH:MM), minimum wind direction (deg), averaged wind direction (deg), maximum wind direction (deg), minimum wind speed ($m·s^{-1}$), averaged wind speed ($m·s^{-1}$), maximum wind speed ($m·s^{-1}$), temperature (°C), relative humidity (%), and pressure (hPa). A summary of the measured values is shown in Fig. 6.

### 3.2 Halo Doppler lidar

A Halo Photonics Stream Line XR scanning Doppler lidar (Pearson et al., 2009) was installed at the location of the weather

station, at the height of 1.3 m AGL. Stream Line XR is capable of full hemispheric scanning and the scanning patterns are





**Table 2.** Technical specification of the Halo Doppler lidar.

| | |
|---|---|
| Wavelength | 1.5 $\mu$m |
| Pulse repetition rate | 10 kHz |
| Nyquist velocity | 20 m·s$^{-1}$ |
| Sampling frequency | 50 MHz |
| Velocity resolution | 0.038 m·s$^{-1}$ |
| Points per range gate | 10 |
| Range resolution | 30 m |
| Maximum range | 12000 m |
| Pulse duration | 0.2 $\mu$s |
| Lens diameter | 8 cm |
| Lens divergence | 33 $\mu$rad |
| Telescope | monostatic optic-fibre coupled |

fully user-configurable. In the vertically pointing mode, the lidar alternates between co- and cross-polar receiver. The minimum range of the lidar is 90 m and its instrumental specifications are given in Tab. 2.

During the campaign at Hailuoto, the scanning schedule included five scans in addition to the vertically-pointing stare with alternating co- and cross-polar measurements. The scans were: (1) a sector scan at 0° elevation angle, azimuth angle ranging from 180° to 360° at 5°-steps; (2) a sector scan at 2° elevation angle, azimuth angle ranging from 180° to 360° at 10°-steps; (3) a vertical azimuth display (VAD) scan at 10° elevation angle, with 15°-steps in azimuth angle; (4) a VAD scan at 70° elevation angle, with 15°-steps in azimuth angle; and (5) a vertically-pointing co-polar scan repeated for 12 rays. The integration time for each scan type was set to 6 s. Sector scans and VADs (scans 1–4) were used to retrieve horizontal winds and a proxy for turbulence at different height and range similar to Vakkari et al. (2015). The last scan (5) was used to estimate turbulent kinetic energy (TKE) dissipation rate according to O'Connor et al. (2010).

The measurements were post-processed according to Vakkari et al. (2019) and the attenuated backscatter ($\beta$) was calculated from signal-to-noise ratio (SNR) taking into account the telescope focus (infinity). The uncertainties in radial velocity and $\beta$ were calculated according to O'Connor et al. (2010). The data were visually inspected and range gate 14 was excluded from further analyses due to increased noise floor. Both the original radial velocity data and the retrieved parameters, i.e., the horizontal wind speed and direction, TKE dissipation rate and turbulence proxy (Vakkari et al., 2015), are stored in data files in netCDF format.

## 4  The HAOS Dataset

For each UAV-UG1 flight listed in Tab.1, two files in the csv format are available with measurements from both meteorological sensors (Bosh BME280) (e.g. 'Flight 1.01-sensor 1' and 'Flight 1.01-sensor 2'), which collected data simultaneously during



**Table 3.** Intercept and slope coefficients for the calibration of UAV-UG1 and UG2 meteorological sensors calibration.

| Temperature | Intercept | Slope |
|---|---|---|
| Sensor 1 | -0.55137±0.04042 | 1.00979±0.00236 |
| Sensor 2 | 0.52777±0.0091 | 0.99421±5.496E-4 |
| Relative Humidity | | |
| Sensor 1 | -3.1004±1.93965 | 1.06163±0.02648 |
| Sensor 2 | -1.42983±1.13164 | 1.08781±0.01615 |

the flight. Each file includes the following variables [description (name in the file)]: geolocation data from GPS sensor: latitude (lat), longitude (lon), UTM coordinates (xUTM, yUTM), altitude (alt), date (date), time (time), date and time in serial date numbers format (t), air pressure (P), temperature (T), relative humidity (RH). The altitude values, due to the high uncertainties in the GPS sensor output, were calculated from the atmospheric pressure $P$ and temperature $T$ measurements using the hypsometric equation:

$$h = \frac{\left( \left( \frac{P_0}{P} \right)^{\frac{1}{5.527}} - 1 \right) (T + 273.15)}{0.0065}, \tag{1}$$

where $P_0$ denotes the surface pressure from the weather station. The initial and final flight segments outside of the target survey path were removed from the files, as described earlier in section 2.3. Apart from this process, no data was rejected and no missing values were found. Example measurements from Flight 3.2 are presented in Fig. 4. The UAV-UG1 and UAV-UG2 measurements of temperature and relative humidity over the survey area are provided without calibration – as they were

measured. The calibrated values of the temperature and relative humidity from both UAV-UG2 sensors can be obtained with a linear calibration equation, y=a+bx, with intercept (a) and slope (b) coeffcients from Table 3. All correlation coefficients (Pearsons's r, R-Square and Adjusted R-Square) are higher than 0.997. Due to the sensors exposure to sunlight dependent on the relative orientation of the aircraft and the sun (different during different fragments of the survey path and changing throughout the day), measurements from the sensor with lower air temperature are recommended for further analysis.

The orthomosaic maps (Fig. 5) of the surface below UAV-UG1 flights path are available in GeoTiff and KML format.

The Halo Doppler lidar dataset consists of 7 netCDF files per day, for each day between 28 February and 2 March 2020. Each file name begins with the prefix 'YYYYmmdd' indicating the day of the measurements and affix related to file contents: (1) co-polar and (2) cross-polar background measurements (co.nc and cross.nc), (3) TKE dissipation rate retrieved from the measurements (TKE.nc) and 4 VAD scans of horizontal wind speed and direction with the elevation angles of (4) 0° (VAD0-

wind.nc), (5) 2° (VAD2-wind.nc), (6) 10° (VAD10-wind.nc), (7) 70° (VAD70-wind.nc). A detailed description of Halo Doppler lidar measurements post-processing can be found in section 3.2.

The automatic weather station measurements are provided in the csv format with a separate file for each day of the campaign. The files, labeled with the prefix 'aws' for automatic weather station and the relevant date, include all the variables listed in section 2.3. The 3D anenometer measurements conducted at the same location are provided in raw, hourly generated, csv

files with the following variables: time in the 'HHMMSS.ss' (hours, minutes, seconds, miliseconds) format; three wind speed components: u, v, w ($10^{-2}$ m·s$^{-1}$) and acoustic temperature $T_s$ ($10^{-2}$ °C).

## 4.1 Data availability

All the described datasets are available to the public in the described formats at https://doi.pangaea.de/10.1594/PANGAEA.918823 (Wenta et al., 2020). The repository is hosted by Alfred Wegener Institute, Helmholtz Center for Polar and Marine
Research (AWI) and Center for Marine Environmental Sciences, University of Bremen (MARUM).

## 5 Summary

During the HAOS campaign, between 27 February and 2 March 2020, 27 fixed wing UAV-UG1 flights were carried out offshore the westernmost point of the Hailuoto island together with overlapping photogrammetry missions which resulted in 4 orthomosaic maps of the sea ice below. Additionally, a 3D sonic anenometer, automatic weather station and Halo Doppler
lidar operated near the Hailuoto Marjaniemi lighthouse throughout the time of the HAOS project.

The primary focus of HAOS was to obtain detailed measurements of the atmospheric boundary layer over sea ice. In accordance with this goal sUAV flights provided continuous 3 dimensional meteorological observations over sea ice offshore and were supplemented by on-shore measurements of atmospheric state. Thus, the presented dataset provides a thorough description of the atmospheric conditions over newly formed sea ice near Hailuoto island. Furthermore, detailed orthomosaic
maps provide a unique and extremely detailed view on the newly formed sea ice and its changes in the span of 4 days (Fig. 5). Considering the scarcity of recent ABL observations over diminishing sea ice cover in the Bay of Bothnia, and the Baltic Sea in general, the presented dataset may be considered as a valuable source of information and the basis for further studies on sea ice-atmospheric interactions in this region. Additionally, as the weather conditions throughout the campaign resembled the ones observed over sea ice in the Arctic, the HAOS dataset can also be used in the studies related to polar regions.

*Author contributions.* D.B. and K.D. designed, constructed and configured the drones. V.V. was responsible for the lidar measurements. M.W. and A.H. planned the research. All authors contributed to the conduction of measurements and data processing, and discussed the results. M.W., D.B. and V.V. wrote the manuscript.

*Competing interests.* The authors declare that they have no conflict of interest.

*Acknowledgements.* This work was funded by the Polish National Science Centre grant No. 2018/31/B/ST10/00195 "Observations and
modeling of sea ice interactions with the atmospheric and oceanic boundary layers".



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
