# Peer review of "Winter atmospheric boundary layer observations over sea ice in the coastal zone of the Bothnian Bay (Baltic Sea)"

_Earth System Science Data, 2020_

## Referee Comment (RC1) · Anonymous Referee #1 · 19 Oct 2020

This is a review of *Winter atmospheric boundary layer observations over sea ice in the coastal zone of the Bothnian Bay (Baltic Sea)* by Wenta et al., submitted to ESSD. The manuscript describes atmospheric boundary layer observations of sea ice in the Bothnian Bay, collected with UAVs over 4 days of 2020 northern hemisphere spring. Overall, the manuscript is well written, concise and the technical details of the flights and measurements are easy to follow. The data was mostly easy to access and consistent with the description in the manuscript. I have some minor comments and suggestions below.

Manuscript:

General comments:

I don't have a feeling for how good NWPs and reanalyses capture the atmospheric boundary layer conditions in this region (or other polar regions with similar characteristics). The authors state that the small-scale processes at the atmosphere-sea ice-ocean interface are crucial for improving NWP models in the polar region. I understand that this dataset can be useful for informing or evaluating NWPs or reanalyses, but has the specific study site been flagged as a particular region of interest in any previous studies, or is it a matter of being a convenient location logistically while also being representative?

I note that no clouds were present below the aircraft during the campaign, but were clouds present above the aircraft during any of the flights? Some of the atmospheric conditions as well as the measurements could be affected by clouds (and vice versa), so this could potentially be useful to indicate (if known).

Minor comments:

Line 37-39. How do the measurements from the HAOS compare to those from the ISOBAR campaign?

Figure 1c. Hailuoto is hard to find on the map and it doesn't appear to be highlighted or distinguished from other labels. If the label could be made bigger or a bounding box around Hailuoto that would be good.

Figure 2. I recommend changing the font of the "open water" to white or a lighter colour since it is too dark to read easily.

Line 94. CMOS isn't defined.

Dataset:

I downloaded the zip file containing all the files from https://doi.pangaea.de/10.1594/PANGAEA.918823. Within that are tab-delimited files containing measurements from the 3d anemometer and the automated weather station from the ground-based measurements, the measurements from both sensors on the UAV, as well as links to download the netcdf files containing the LIDAR data.

Using the R packages readr and tidync, I read in a sample of each different type of measurement to inspect the data and produce some quick plots. I also did a quick comparison of the altitude data from the two UAV sensors during flight 1, which appeared mostly consistent with each other. The files in the zip folder are ".tab" format and are tab delimited, not .csv as described in the manuscript. Aside from this, I had no problems reading the data and everything else appeared consistent with

what was described in the manuscript. Note that for the LIDAR netcdf files, I only checked one of the co-polarisation data files.

I was not able to download the orthomoasic map data. With my logged in Pangea account, I get the following error: "Your client is not allowed to access the requested object. This may because you are logged in with the wrong user account." I created an account on Pangea for the first time to download this dataset, so I'm not sure if it is something to do with Pangaea or the permissions set by the authors of the dataset.

---

## Referee Comment (RC2) · Anonymous Referee #2 · 30 Oct 2020

This review addresses the manuscript Winter Atmospheric Boundary Layer Observations over Sea Ice in the Coastal Zone of the Bothnian Bay (Baltic Sea) by Wenta et. al. The authors introduce a dataset of the atmospheric boundary layer of sea ice in Bothnian Bay that was obtained from aerial observations. The description of the dataset is well done, I have very few comments/corrections, and I recommend this manuscript to be accepted pending minor revision. Below my comments, I include general comments first, followed by correction recommendations by line number:

General

Justify the study area a little more. On line 38, you mention this study site was also the
location of ISOBAR, so this area must have some significance for study in a broader context. I would be good to explain that in your manuscript. Is it just a good place to test this method out on? If there are other factors making this area a good study location, it would be good to articulate for other people hoping to replicate your methods.

Figure 1: In the scaled-up map of Finland, it would be good to make the location of Marjaniemi harbor easier to identify. Perhaps a large colored symbol that can catch the eye better. I had to look at the figure for a little while before figuring out where it is.

Line-byline

Lines 1-12: The Abstract could use a line or two summarizing the importance of this data and its possible uses, especially in relation to your selected study area. The introduction does this fairly well, and I think even copy/pasting a few sentences from that section into the abstract would work fine.

Line 23: Perhaps give an example or two of physical processes studying ABL properties helps us to understand. Could even be in parenthesis (e.g. property A, property B).

Line 180: Could you provide a little more information on the orthomosaic maps for this section? map extent, resolution, etc, to match the detail you gave to the other datasets described in this section.

Please also note the supplement to this comment:
https://essd.copernicus.org/preprints/essd-2020-153/essd-2020-153-RC2-supplement.pdf

———————————————————

---

## Author Comment (AC1) · 5 Nov 2020

In the first place we would like to thank the Reviewers for their comments and suggestions, We are thankful for the insightful analysis of our study, as it will certainly make the paper more comprehensible for future readers.

Reviewer 1

This review addresses the manuscript Winter Atmospheric Boundary Layer Observations over Sea Ice in the Coastal Zone of the Bothnian Bay (Baltic Sea) by Wenta et. al. The authors introduce a dataset of the atmospheric boundary layer of sea ice in Both-

nian Bay that was obtained from aerial observations. The description of the dataset is well done, I have very few comments/corrections, and I recommend this manuscript to be accepted pending minor revision. Below my comments, I include general comments first, followed by correction recommendations by line number:

General Justify the study area a little more. On line 38, you mention this study site was also the location of ISOBAR, so this area must have some significance for study in a broader context. I would be good to explain that in your manuscript. Is it just a good place to test this method out on? If there are other factors making this area a good study location, it would be good to articulate for other people hoping to replicate your methods.

In fact, we do provide the justification of our choice of study location in the Introduction section, but we agree that the reasons have not been formulated clearly enough. In the revised manuscript, we add a clearly formulated list of reasons which, apart from practical aspects like an easy access by car, availability of electricity at the pier of the Hailuoto harbor that enabled the installation of the weather station, etc., includes a very important fact that the drifting ice pack of interest in this study was present within a few hundred meters from the shore, i.e., within reach of our UAVs. This was absolutely crucial for our campaign, otherwise we would have had access to fast ice only. As far as the similarities and differences between ISOBAR and HAOS are concerned, we added a short information about the main similarities and differences between them (we also added reference to the new, very recent paper describing ISOBAR).

Figure 1: In the scaled-up map of Finland, it would be good to make the location of Marjaniemi harbor easier to identify. Perhaps a large colored symbol that can catch the eye better. I had to look at the figure for a little while before figuring out where it is.

We modified the figure according to the reviewer's suggestion.

Line-byline Lines 1-12: The Abstract could use a line or two summarizing the importance of this data and its possible uses, especially in relation to your selected study

area. The introduction does this fairly well, and I think even copy/pasting a few sentences from that section into the abstract would work fine.

The abstract has been changed according to reviewer's comments.

Line 23: Perhaps give an example or two of physical processes studying ABL properties helps us to understand. Could even be in parenthesis (e.g. property A, property B).

The examples have been added to the sentence.

Line 180: Could you provide a little more information on the orthomosaic maps for this section? map extent, resolution, etc, to match the detail you gave to the other datasets described in this section.

We added information about the spatial resolution of each map to the text. The information about the extent of the maps is provided in the caption of Fig.5, which we extended to make it more precise.

Reviewer 2

This is a review of Winter atmospheric boundary layer observations over sea ice in the coastal zone of the Bothnian Bay (Baltic Sea)by Wenta et al., submitted to ESSD. The manuscript describes atmospheric boundary layer observations of sea ice in the Bothnian Bay, collected with UAVs over 4 days of 2020 northern hemisphere spring. Overall, the manuscript is well written, concise and the technical details of the flights and measurements are easy to follow.The data was mostly easy to access and consistent with the description in the manuscript. I have some minor comments and suggestions below.

Manuscript: General comments:

I don't have a feeling for how good NWPs and reanalyses capture theatmospheric boundary layer conditionsin this region(or other polar regions with similar characteristics).The authors state that the small-scale processes at the atmosphere-sea ice-ocean interface are crucial for improving NWP models in the polar region.I understand that this dataset can be useful for informing or evaluating NWPsor reanalyses, but has the specific study site been flagged as a particular region of interest in any previous studies, or is it a matter of being a convenient location logisticallywhile also being representative?

As we already wrote in our reply to reviewer #1, the main reason for our choice of study location – apart from practical arguments related to logistics – was the special position of the Marjanemi harbour as an exposed, westernmost point from which the UAVs could access the drifting ice pack (as opposed to fast ice, covering a wide zone along the coast elsewhere). We added information on that to the revised manuscript – or rather, formulated the information that was available in the first version of the manuscript in a more clear form.

I note that no clouds were present below the aircraft during the campaign, but were clouds present above the aircraftduring any of the flights? Some of the atmospheric conditionsas well as the measurementscould be affected by clouds(and vice versa), so this could potentially be useful to indicate(if known).

A new sentence describing the weather conditions throughout the campaign have been added to the section "Mission Planning".

Minor comments: Line 37-39. How do the measurements from the HAOS compare to those from the ISOBAR campaign?

We added a new text briefly describing the differences between the two campaigns. However, we do not include any in-depth analysis of the two datasets, as this is part of a follow up research that is under preparation.

Figure 1c. Hailuoto is hard to find on the map and it doesn't appear to be highlighted or distinguished from other labels. If the label could be made bigger or a bounding box around Hailuoto that would be good.

We modified the figure according to the reviewer's suggestion.

Figure 2. I recommend changing the font of the "openwater"to white or a lighter colour since it is too dark to read easily.

We modified the figure according to the reviewer's suggestion.

Line 94. CMOS isn't defined.

We added an explanation of the CMOS acronym.

Dataset: I downloaded the zip file containing allthe files from https://doi.pangaea.de/10.1594/PANGAEA.918823. Within that aretab-delimited files containing measurements from the 3d anemometerand the automated weather station from the ground-based measurements, the measurementsfrom both sensors on the UAV, as well as links to download the netcdf files containing the LIDAR data. Using the R packages readr and tidync, I read in a sample of each different type of measurement to inspect the dataand produce some quick plots. I also did a quick comparison of the altitude datafrom the two UAV sensors during flight 1, which appeared mostly consistent with each other. The files in the zip folder are ".tab"format and are tab delimited, not .csv asdescribed in the manuscript. Aside from this, I had no problems reading the data and everything elsea ppeared consistent with what was described in the manuscript. Note that for the LIDAR netcdf files, I only checked one of the co-polarisation data files.

We made corrections in the manuscript, changing 'csv' to 'tab delimited'.

I was not able to download the orthomoasic map data. With my logged in Pangea account, I get the following error: "Your client is not allowed to access the requested object. This may because you are logged in with the wrong user account."I created an account on Pangea for the first time to download this dataset, so I'm not sure if it is something to do with Pangaea or the permissions set by the authors of the dataset.

I managed to successfully download the Orthomosaic data today, so I presume that

there was a problem with PANGAEA database. We didn't set any special permissions required to access the orthomosaic map data, all data included in our dataset are freely available to everyone.

Thank you for checking the data availability and consistency – it is good to know from an independent person that everything is there and works fine!

———————————————

**(a)**

LEGEND
- Flight 2.04*
- Flights 1.1-4.3
- Flight 0.2, Flight 0.3
- Flight 0.1
- Weather station and Halo Dopper Lidar
- Orthomosaic maps area
- First orthomosaic map (Day 0)

0    250    500 m

**(b)**

~ 1,37 km

1
~ 1,3 km
~ 0,35 km
4
3
~ 1,1 km
6
5
~ 1,1 km
7

Take off and landing spot.

**(c)**

Luleå
Halluoto
Oulu
Vaasa
Sundsvall
SWEDEN
FINLAND
Gulf of Bothnia
Turku
Helsinki
Åland Islands
Gulf of Finland
St. Petersburg
Stockholm
Tallinn
Hiiumaa
ESTONIA
RUSSIA
Saaremaa
Gulf of Riga
Gotland
Baltic Sea
Öland
LATVIA
Riga
LITHUANIA
Vilnius
Bornholm
Kaliningrad (RUSSIA)
Minsk
Gdansk
BELARUS
Rügen
POLAND

**Fig. 1.**

[Figure]

fast ice

ice
pack

open
water

**Fig. 2.**